# Balancing the supply and demand for taxonomy: An analysis of European taxonomic capacity and policy needs

Quentin J. Groom[1]*, Melanie De Nolf[1], Lina M. Estupinan-Suarez[2,3], Sofie Meeus[1]

1 Meise Botanic Garden, Meise, Vlaams-Brabant, Belgium, 2 German Centre for Integrative Biodiversity Research (iDiv), Halle-Jena-Leipzig, Saxony, Germany, 3 Institute of Biology, Martin Luther University Halle-Wittenberg, Halle, Germany

* quentin.groom@plantentuinmeise.be

## Abstract

Taxonomy is a cornerstone of biological science and essential to biodiversity policy, yet it faces persistent structural challenges collectively known as the "taxonomic impediment". These include limited capacity, uneven geographic and taxonomic coverage, and a disconnect between the supply of expertise and its societal demand. In this study, we present a meta-research analysis of taxonomic activity in Europe over the past decade, drawing on publication metadata from OpenAlex, Wikidata, and GBIF. Using an open and reproducible workflow, we identify more than 31,000 authors affiliated with European institutions who have contributed to taxonomic publications, and we assess their taxonomic and institutional distribution. Using robust regression models, we show that biodiversity policy variables collectively explain additional variation in taxonomic research effort beyond species richness alone, with the Birds and Habitats Directives showing positive associations and marine-related policy variables showing negative associations. We explore how this supply of expertise compares with demands arising from European biodiversity policy, including legally binding instruments such as the Birds and Habitats Directives and the Marine Strategy Framework Directive, as well as strategic initiatives focused on invasive alien species, crop wild relatives, and species of conservation concern. Our results highlight clear imbalances in capacity across taxonomic groups and regions, with some politically and ecologically significant taxa receiving comparatively little attention. This work illustrates how openly available data can be used to evaluate taxonomic capacity and its alignment with policy needs, providing a framework for strategic planning and investment in taxonomy.

## 1 Introduction

Every day, species are disappearing from our planet at an alarming rate, yet our understanding of even the most basic building blocks of life remains incomplete. The

**Data availability statement:** The source code for the workflow and the data used are available on GitHub (https://github.com/AgentschapPlantentuinMeise/TETTRIs-mapping-taxonomists) and the latest archived release is deposited on Zenodo (https://doi.org/10.5281/zenodo.16356773) and made available under a CC-BY 4.0 license (De Nolf et al. 2025). The European border shapefiles used for map generation were obtained from Natural Earth (https://www.naturalearthdata.com), a public domain dataset. Bibliographic data are retrieved from OpenAlex (https://openalex.org/). Taxonomic information and biogeographic information were retrieved from the Global Biodiversity Information Facility (https://www.gbif.org/).

**Funding:** MDN and SM were supported by the TETTRIs project, which receives funding from the European Union's Horizon Europe Innovation Actions (https://research-and-innovation.ec.europa.eu/funding/funding-opportunities/funding-programmes-and-open-calls/horizon-europe_en) under grant agreement No. 101081903 and by the DiSSCo Flanders project, funded by the Research Foundation – Flanders (FWO) research infrastructure under grant number I001721N. QG and LES were supported by the B3 project (Biodiversity Building Blocks for Policy), funded by the European Union's Horizon Europe Research and Innovation Programme (https://research-and-innovation.ec.europa.eu/funding/funding-opportunities/funding-programmes-and-open-calls/horizon-europe_en) under grant agreement No. 101059592. The funders had no role in study design, data collection and analysis, decision to publish, or preparation of the manuscript.

**Competing interests:** The authors have declared that no competing interests exist.

"taxonomic impediment" is widely recognised, including by the Convention on Biological Diversity as a barrier to achieving biodiversity policy and management objectives because the taxonomic knowledge, infrastructure, and expertise needed to identify, monitor, and manage species are insufficient or unevenly available [1]. This impediment encompasses the challenge of documenting and cataloguing Earth's biodiversity, but also the broader limitations on translating taxonomic knowledge into conservation assessment, monitoring, and legally binding decision-making. The concept has been frequently discussed in relation to the various challenges that constrain taxonomic capacity, including limited funding, a shortage of expert taxonomists, technological constraints, restricted data accessibility, and a lack of global collaboration and standardised methodologies [2–4].

Taxonomy is a fundamental discipline of biology, it is essential for detailed communication about the diversity of life. Taxonomists possess a wide range of skills, from applying nomenclature codes and identifying specimens to evaluating traits through microscopy, chemistry, and genetics. They are involved in teaching, identification, monitoring biodiversity, and generating new knowledge on evolution, form, and function [5,6]. Effective investment in taxonomy requires balancing the study of certain taxa, the discovery of new species, the focus on particular issues (e.g. invasive alien species), and the need for sound taxonomic foundations for applied research and conservation.

In this context, taxonomists have long complained about the evaluation of their work by the use of publication citation and Impact Factors that fail to capture the true scientific and societal impact of their research [2,7,8]. Scientific citation practices fail to pick up the impact of describing new species and fail to recognise the long shelf life of taxonomic research in comparison to the other sciences. Yet while taxonomists believe they are underappreciated, there seems to be little self-reflection by taxonomists on what impactful taxonomy really is: which taxa are most impactful to study, and what outputs of taxonomy are most useful to science and society? Public funders typically prioritise societal and political needs when issuing calls for research funding, and institutions likely align their decisions on recruitment, discretionary funding, and infrastructure investment with these perceived needs [9]. These priorities can be national, such as the development of a national biodiversity strategy, or global, in response to frameworks like the Global Biodiversity Framework [10].

From a public policy standpoint, species are not equally important [11]; decisions regarding funding and conservation efforts often prioritise species that directly impact human wellbeing, such as those that cause or carry disease, or those that provide food, energy, or other ecosystem services. The prioritisation of societal needs by public funders therefore implicitly suggests that a "gap-filling" approach—the idea that taxonomy should aim to describe every species on Earth, irrespective of the potential impact or usefulness of that knowledge—may not always align with strategic funding priorities. Alternatively, frameworks such as Sabatier's Advocacy Coalition Framework (ACF) [12] and the concept of co-production [13] suggest a dynamic, two-way interaction between research and policy, an interaction that funders are increasingly trying to support [14]. Research is strategically used by competing coalitions to shape

policy agendas, while policy priorities, in turn, influence the research deemed relevant and fundable. This is paralleled in the case of scientific initiatives involving the general public and invasive alien species. In this case a beneficial cycle can be created whereby community-based data collection can directly inform policy decisions and policy needs can drive public participation in scientific research [15].

There are numerous demands for taxonomy across various sectors of society and policy, making it challenging to evaluate all of them comprehensively. In this study, we focused on the taxonomy needs that are relevant across Europe, aiming to provide insights for countries within the European Union, as well as neighbouring and affiliated nations. However, we recognize the diverse national demands for taxonomy, which can vary significantly depending on the predominant habitats within each country—such as marine, alpine, freshwater, peatland, and karst environments. Additionally, these demands are influenced by historical connections to former colonies and their biodiversity, current links to overseas territories, as well as by key industries like agriculture, forestry, and fisheries, and local environmental challenges such as pollution [16,17].

We particularly focus on policy areas with a clear need of taxonomic services, invasive alien species, crop wild relatives, conservation worthy species and specific policy instruments of the European Union. For example, Since 2014 the European Union has legislated against a blacklist of invasive species that bans the trade and mandates the control of these species. While taxonomic knowledge is needed to positively identify these species, their names, and their scope, there are a much larger number of species that may become established in European countries, and for which taxonomic expertise is needed to ensure rapid detection and potential eradication. For this reason we also used a list of species created in a horizon scanning exercise that used expert opinion to identify species that show invasiveness on other continents and may become established in Europe [18].

The IUCN Red List of Threatened Species (hereafter referred to as the Red List) is a critical tool for conservation policy, serving as a comprehensive repository of species assessments that inform and shape conservation strategies worldwide [19,20]. With its detailed, species-level insights into the five primary drivers of biodiversity loss, the Red List is invaluable for policy initiatives like the EU Biodiversity Strategy for 2030, which seeks to halve the number of Red List species impacted by invasive alien species [21].

Crop wild relatives (CWR) are recognised as an irreplaceable genetic resource for the improvement of crops [22]. Effective policies for crop wild relatives should promote their conservation, establishing national, regional, and global information systems, and developing mechanisms to prioritise conservation efforts, and should promote the integration of CWR conservation and other conservation of plant genetic resources.

There are also specific legal instruments of the European Union such as the Birds Directive (Directive 2009/147/EC) which is one of the cornerstone legislative frameworks for biodiversity conservation in the European Union. It aims to protect all wild bird species naturally occurring in Europe by safeguarding their habitats, regulating hunting, and establishing Special Protection Areas. The directive relies heavily on accurate species identification and taxonomic clarity to inform conservation measures and reporting obligations.

The Habitats Directive (Directive 92/43/EEC) complements the Birds Directive by focusing on the conservation of natural habitats, wild fauna, and flora across Europe. It underpins the Natura 2000 network of protected areas, requiring regular monitoring and assessment of species and habitat conditions. Taxonomic expertise is essential to the directive's implementation, particularly for less well-known taxa such as plants, invertebrates, and fungi listed in the annexes.

The Marine Strategy Framework Directive (Directive 2008/56/EC) provides a framework for the protection of the marine environment across Europe. It aims to achieve Good Environmental Status of the EU's marine waters, requiring comprehensive monitoring of marine biodiversity. Reliable taxonomy is critical for assessing species diversity, detecting invasive species, and evaluating ecosystem health indicators.

More recently the EU Pollinators Initiative addresses the decline of wild pollinating insects, especially bees, hoverflies, and butterflies. While not a directive, it is an integral part of the EU Biodiversity Strategy for 2030 and influences

agricultural policy and funding calls. Its success depends on taxonomic accuracy to monitor pollinator populations, inform conservation actions, and support citizen science engagement.

Addressing these challenges, this study examines policy areas with a clear need for taxonomic services, including invasive alien species, crop wild relatives, and conservation-worthy species, within the framework of EU policy instruments. Building on the recommendations from the European Red List of Insect Taxonomists [23], we broaden both the taxonomic and geographic scope, implementing an open and reproducible workflow to monitor capacity. While the 'taxonomic impediment' has been discussed extensively, there are relatively few reproducible, data-driven assessments that quantify taxonomic capacity at continental scale and test its alignment with major policy instruments. Unlike previous studies that focus solely on taxonomic capacity, our study links publication-based indicators of expertise with multiple explicit policy-demand datasets and uses statistical modelling to assess whether the distribution of taxonomic effort corresponds with these policy signals.

We assessed the supply of taxonomists by analysing the authorship of taxonomic literature. To do this, we developed an automated workflow that draws data from the APIs of three open sources:

- OpenAlex, a database that provides comprehensive metadata about academic publications, authors, institutions, and research topics to facilitate scholarly discovery and analysis.

- Wikidata, a collaboratively edited knowledge base that provides structured data to support Wikimedia projects, and beyond, enabling data-driven applications and research.

- The Global Biodiversity Information Facility (GBIF), an international network and data infrastructure that provides access to data on observations of all types of life on Earth.

The objective of this study is to spark meaningful discussion about the future of taxonomic research and how it is prioritised. We aim to inspire dialogue on the evaluation and monitoring of taxonomy, emphasizing the importance of balancing the supply and demand for taxonomic expertise in planning, training, and recruitment for the field's future, informing effective resource allocation in a politically driven landscape. We also encourage others to replicate our work in their own countries or regions, to compare and contrast their findings with ours, contributing to a more evidence-based and politically informed approach to biodiversity conservation.

## 2 Materials and methods

### 2.1 Policy relevant data

Crop wild relatives data for Europe were downloaded from the database of the Germplasm Resources Information Network online database [24]. IUCN Red List of Threatened Species were downloaded from their online database [20] selecting taxa for Europe and status of *Taxonomic Research Needed*. The full European Redlist was sourced from the European Environment Agency [25]. Importantly, the European Red List programme has assessed only selected taxonomic groups; therefore, taxa absent from these datasets are often unassessed rather than unthreatened, and 'missing' groups should be interpreted primarily as gaps in assessment coverage.

Invasive alien species on the horizon for Europe were sourced from the supplementary data of Roy et al. [18]. The importance of taxonomic capacity for invasive species prevention and management has been highlighted in earlier taxonomic-needs assessments and capacity evaluations, particularly for early detection and rapid response programs [26,27]. The list of European pollinator species was taken from Reverté et al. [28]. Species pertinent to the Marine Strategy Framework Directive were taken from Palialexis et al. [29].

A primary challenge encountered during the analysis of EU biodiversity legislation was the decentralised nature of information. Key data are distributed across various web portals, including those of the European Environment Agency (EEA), the European Nature Information System (EUNIS), and other organisations. Consequently, retrieving essential

information, such as the lists of species covered by the Nature Directives, was a time-intensive undertaking. Furthermore, these sources are often not machine-readable and lack consistent identifiers that link to other taxonomic resources, complicating automated data integration. As a result, these data are often difficult for web browsers to identify and are not consistently available in readily usable tabular formats. To facilitate reproducibility and further analysis, species lists used in the subsequent analyses, along with their associated taxonomy, have been compiled into tabular format and made publicly available on Zenodo [30]. This limitation also constrains automated assessment of the policy relevance of taxonomic outputs at scale, because policy instruments rarely reference persistent taxonomic identifiers in ways that can be programmatically linked to bibliographic databases.

## 2.2 A workflow to identify European authors of taxonomic articles

The workflow we have created, firstly identifies taxonomic journals and extracts articles published in them from the last 10 years. Secondly, it uses keywords in the title and abstract to identify those papers that are considered to be the work of taxonomists, whether or not this was the main role of the author(s), or a subsidiary one. The resulting corpus of taxonomic literature was then used to identify the institutional affiliation of the authors, restricting it to institutions based in Europe, and their taxonomic focus (Europe refers to the European Political Community plus Vatican City, and the dependencies/territories of European countries that themselves are in Europe). Automatic disambiguation of authors with manual validation allowed us to avoid double counting of authors. The workflow is largely written in Python (version 3.9.18) and made available on Zenodo [31]. Though to take advantage of the metacoder package (version 0.3.7) heattree.R was written in R (version 4.3.2) [32]. Additional R packages include gridExtra (2.3) and rcartocolor (2.1.1). The key Python packages required for these scripts to run are pandas (2.1.4), numpy (1.26.4), requests (2.31.0), geopandas (0.12.2), matplotlib (3.9.2), seaborn (0.12.2), SPARQLWrapper (2.0.0), fiona (1.8.22), shapely (2.0.1).

**2.2.1 Taxonomic journal selection.** We used three methods to find journals that could contain taxonomic articles, from two sources: Wikidata and OpenAlex. We used the *requests* Python package to access their APIs. Wikidata provides structured data by linking entities with unique Q numbers and their attributes or relationships using P numbers for properties. Using list_journal.py we searched for all the scientific journals (Q5633421) or academic journals (Q737498) in Wikidata with properties 'main subject' (P921) or 'field of work' (P101) linked to the items 'taxonomy' (Q8269924), 'biological classification' (Q11398), 'plant taxonomy' (Q1138178), 'animal taxonomy' (Q1469725), 'systematics' (Q3516404), 'biological nomenclature' (Q522190), 'botanical nomenclature' (Q3310776), 'zoological nomenclature' (Q3343211), 'phylogenetics' (Q171184) or 'animal phylogeny' (Q115135896).

Similarly, we downloaded all journal records from Wikidata with a property of 'IPNI publication ID' (P2008), or 'ZooBank publication ID' (P2007), meaning any journal with any of those IDs attached. Note that although IPNI is a resource on the names of vascular plants, the bibliographic details from IPNI also contain details of bryological, mycological and phycological journals.

In addition, we searched OpenAlex for all journals, referred to more broadly as 'sources' in OpenAlex, that are associated with the concept 'taxonomy' (C58642233) [33]. In OpenAlex each work is tagged with multiple concepts, based on the title, abstract, and the title of its host venue using an automated classifier that was trained on Microsoft Academic Graph's corpus [33]. Concepts for sources are generated from the most frequently applied concepts to works hosted by this source.

For each journal we retained the display name (title), Wikidata ID, OpenAlex ID, ISSN, ISSN-L, IPNI publication ID, ZooBank publication ID, dissolved status and dissolved year. Journals found through different methods but with the same Wikidata ID and OpenAlex ID, were deduplicated. In the downstream analysis, the journals were accessed via OpenAlex, so any journal that lacked an OpenAlex ID could not be used. The Jupyter notebook Journals.ipynb provided can be used to examine the sources of the journals we discovered [34].

 

**2.2.2 Article selection.** We used the OpenAlex API to request European articles from the preselected journals, published between 2014 and 2023 inclusive. This meant searching for the OpenAlex journal ID using the *primary_location. source.id* filter, selecting articles written by at least one author affiliated to an institution located in these European countries using the *authorships.countries* filter and a list of European two-letter country codes, and setting the begin and end dates to 2014-01-01 and 2023-12-31 in the configuration file.

Articles were filtered for specific keywords to extract taxonomic articles. We selected these words through an interactive refinement process. This involved two main steps: identifying and excluding articles unrelated to taxonomy, and finding and verifying that relevant articles were included. This keyword filter searches the title and abstract for words such as *taxonomic, taxon, checklist, nov.* (only in abstract), *new species, novel species, new genus*, and *new genera* and their translations in Bulgarian, Czech, French, German, Hungarian, Italian, Polish, Portuguese, Romanian, Russian, and Spanish. It also searched the concepts OpenAlex has associated with the article: taxonomy (C58642233) and taxon (C71640776). Finally, all articles were filtered to ensure they were part of the OpenAlex Life Sciences domain (https://openalex.org/domains/1).

**2.2.3 Taxonomic processing.** In 'parse_taxonomy.py', the filtered taxonomic articles were parsed for species names. Specifically, the script uses regular expressions to search the title and abstract for word groups that are capitalised like "*Genus species*". Possible species names were matched to the GBIF taxonomic backbone [35]: if the word group matched one of the species names found in the backbone, it was saved and added as metadata to the article. Additionally, the rest of the text was searched for other species of the same genus like "*G. species*" and again matched to the backbone.

**2.2.4 Author processing and disambiguation.** OpenAlex provides a list of "authorships" for each article, containing for each author their name, IDs, institutions mentioned in the article, countries where they work, and much more. 'get_ authors.py' extracts the authors from the articles dataframe. This list of authors is deduplicated based on the OpenAlex author ID. The algorithm finds articles with at least one European author, but they may have collaborated internationally, so the authors are filtered to only include those with European affiliations.

Although OpenAlex usually correctly assigns a single author ID to distinct individuals, it sometimes fails to detect duplicates, assigning multiple IDs to the same person. We resolved ambiguities among possibly duplicated authors from taxonomic articles by considering their names, affiliations and taxonomic group of expertise. We created a dataset of authors and generated simplified versions of their names: one consisting of their stripped names with spaces, periods and hyphens removed, and one consisting of their first initial and last name.

We then checked all authors with the same truncated name. Two authors are matched and considered to be the same person if one of two cases is true: either they have the exact same stripped name and work at the same institution or they have the same truncated name, work at the same institution and work on the same taxonomic orders. If one of these matches occurs the information from the two author entries are then merged to create a unified record. The final disambiguated list is saved for further analysis. Some authors (5% before deduplication) study taxa without order rank assigned in the GBIF taxonomic backbone. To disambiguate them we used the family rank. The number of taxonomists was correlated with national population size sourced from the World Bank [36].

## 2.3 Collectors of specimens in Europe

Individuals who collected specimens are listed in the Darwin Core *recordedBy*-field, while those who determined the taxonomic identity of the specimens are listed in the *identifiedBy*-field. If this is the same person as mentioned in the *recordedBy* term the same name may be used in the *identifiedBy*-field, or this field is often left blank. These fields contain names of people as an uncontrolled text string that may contain the name of an individual, an organisation, expedition, or multiple individuals. We use a count of the unique *recordedBy* and *identifiedBy* as an indication of the number of actual people involved in recording and identifying biodiversity in a country.

Aggregated counts of distinct *recordedBy* and *identifiedBy* fields were extracted from GBIF using a SQL command querying the occurrence data to calculate, for each European country, the total number of records, the number of distinct observers (*recordedBy*), and the number of distinct identifiers (*identifiedBy*). We included only records with an *occurrenceStatus* of "PRESENT", excluded records flagged with the "COUNTRY_COORDINATE_MISMATCH" issue, and restricted the dataset to the years 2014–2023. The results were grouped by country (*countryCode*) and sorted in ascending order.

## 2.4 Analysis

To investigate the influence of biodiversity policies and species richness of orders on the number of authors involved in taxonomic research, we conducted a robust statistical analysis [37]. The predictor variables are the number of named species in each order of plants, fungi and animals, and the number of species named in each policy. We began by log-transforming both predictor variables (policies and species richness of orders) and the response variable (number of authors) using the natural logarithm ($\ln(x + 1)$) to stabilise variance and improve normality.

Numbers of described species in each taxonomic order were calculated from the GBIF Taxonomic Backbone [35]. This was done by counting the number of accepted species in the Backbone for each order.

We fitted two regression models using Robust Linear Models (RLM) with Huber's T norm to mitigate the influence of outliers: 1) A combined model that included species richness and policy-related variables: 'taxonomicResearchNeeded', 'cropWildRelatives', 'iasListConcern', 'horizonInvasives', 'habitatsDir', 'marineDir', 'redlistFull', 'birdDir', and 'pollinators'. 2) A reduced model including only species richness. To assess the significance of policy predictors collectively, we compared the two models using an F-test. We evaluated model assumptions by visually inspecting residuals through scatterplots and Q-Q plots and conducting a Shapiro-Wilk normality test on residuals (S6, S7, S8, S9 Figs).

Finally, we identified the top residual outliers from the robust model to explore taxa that deviated notably from the model predictions, potentially highlighting additional unmeasured factors influencing research effort. The Python code for the statistical analysis can be found in [31].

# 3 Results

## 3.1 Demands from biodiversity policy for taxonomists

### 3.1.1 Species of conservation concern.
Of the 13,918 European species on the Red List, 13.4% (1,866 species) have been classified by assessors with the status of *Taxonomic Research Needed* indicating issues with species delimitation. Among these, 4.5% are critically endangered (83 species), 8.1% endangered (152 species), and 10.3% vulnerable (193 species) [20]. Additionally, 13.3% (1850 species) of the European species on the Red List are classified as *Data Deficient*. A data deficient species lacks sufficient information for a proper conservation assessment, often due to a lack of data on its abundance, distribution, or taxonomy.

Taxonomically, the plant species in the *Taxonomic Research Needed* classification are primarily from Tracheophyta (Magnoliopsida, Liliopsida), and Bryophyta, with minimal representation of algal groups (Fig 1). In the case of animals, Gastropoda, Insecta, and Actinopterygii are the key groups, while classes like Clitellata, Echinodermata, and Platyhelminthes are not covered or only sparsely covered in the European Red List assessments currently available, reflecting a bias toward vertebrate, plant, and selected insect groups (Fig 1). Few fungal species (*N* = 77) are in the *Taxonomy Research Needed* category, and these are largely from the Agaricomycetes. Similarly, the EU Biodiversity Strategy for 2030's European Red List of Species highlights plants mainly from Magnoliopsida and animals primarily comprising vertebrates, Gastropoda, and Insecta (Fig 1, Table 1).

### 3.1.2 European crop wild relatives list.
Five hundred and twenty-four species of crop wild relatives are listed for Europe [24]. The most species-rich genera are listed in S1 Table which consist of 43% of all crop wild relatives and

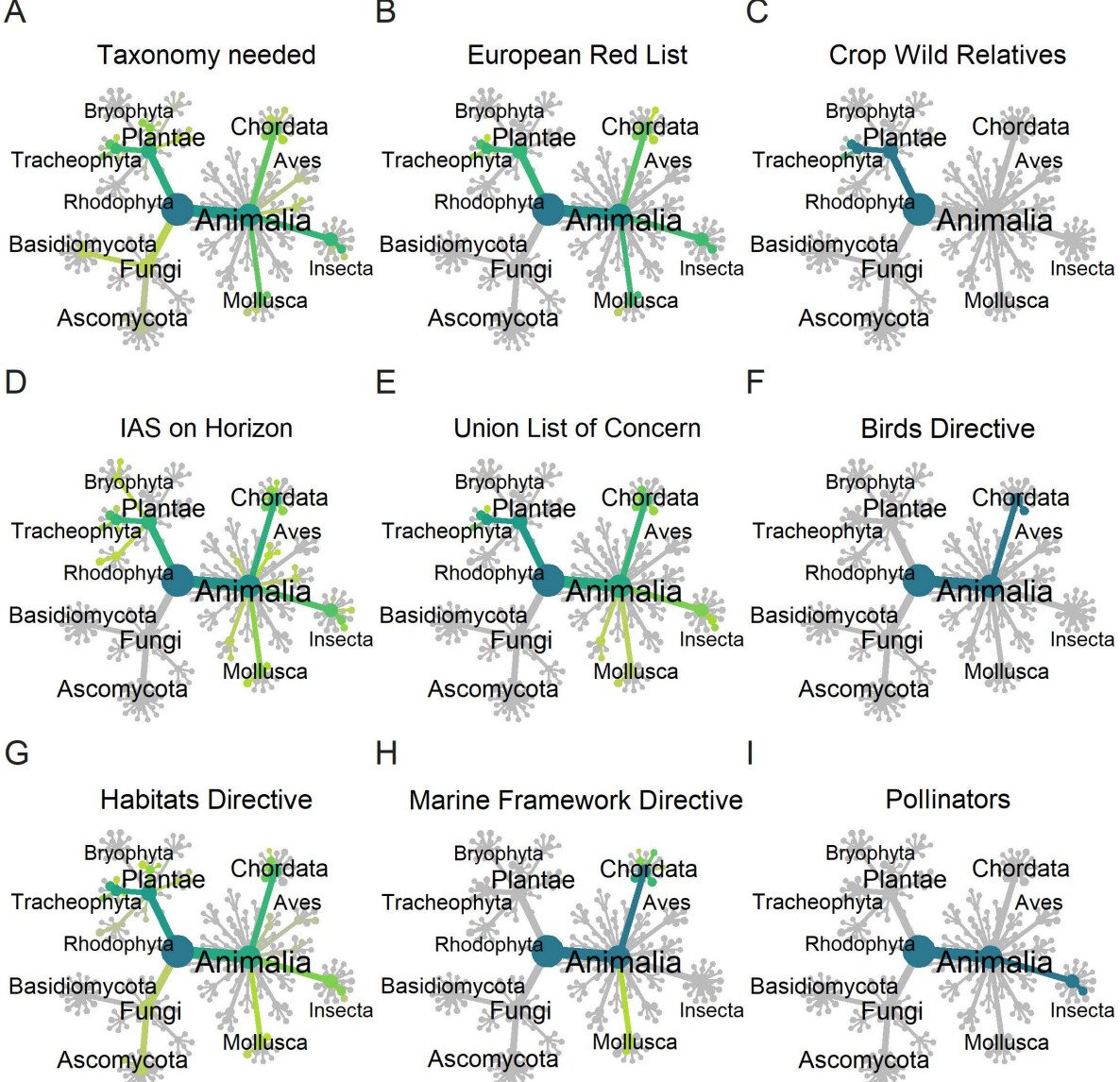

**Fig 1. Phylogenies of the number of species in different policy relevant categories.** Number of species in **(A)** the taxonomic research-needed category in IUCN Redlists; **(B)** the European Red List; **(C)** the European crop wild relatives; **(D)** the invasive alien species on the horizon for future invasions; **(E)** the Union List of Concern for invasive species; **(F)** the species named in the European Birds Directive; **(G)** the species named in the European Habitats Directive; **(H)** the species named in the European Marine Framework Directive and **(I)** the species important to the European Pollinator Initiative. The colour gradient in the heat trees ranges from yellow (low values) to blue (high values), with grey denoting the absence of species.

these come from only five families Fabaceae, Brassicaceae, Poaceae, Rosaceae and Amaryllidaceae. Among those are relatives of staples like wheat, forage crops like alfalfa, oil crops like rape and fruits, such as peach. Therefore, even within the vascular plants these plants are narrow in taxonomic scope.

**3.1.3 Invasive alien species.** Roy et al. [18] identified 120 species in a horizon scan for invasive species in Europe, and these were ranked from medium to very high likelihood of establishment (Fig 1). Among plants these are mainly in

**Table 1. Number of species covered by selected EU legislation and policy instruments. ~ denotes approximate values due to incomplete data or taxonomic uncertainties. Some species may be listed under multiple legislative acts.**

| Legislation | Species | Taxonomy | Reference |
| --- | --- | --- | --- |
| Birds Directive (Annex I) | 193 | Aves | The Council of European Communities (1979) [38] |
| Habitats Directive (Article 17 checklist, 2020) | ~1510 | Animals, fungi and plants | The Council of European Communities (1992) [39] |
| Marine Water Framework Directive (Descriptor 1) | 368 | Mostly vertebrates, such as fish, birds, and marine mammals and reptiles | Palialexis & Boschetti (2021) [29] |
| IAS list of Union Concern (2020) | 88 (41 plants, 47 animals) | The animals are mostly vertebrates, such as fish, birds, and mammals, and the plants are mostly Magnoliopsida and Liliopsida | European Parliament, Council of the European Union (2019) [40] |
| Pollinators Initiative | 3051 | Bees (Anthophila) and hoverflies (Syrphidae) | European Commission, Directorate-General for Environment (2021); Reverté et al. (2023) [21,28] |
| EU Biodiversity Strategy for 2030 / European Red List (VU or higher) | 1864 | The animals are mainly vertebrates, Gastropoda and Insecta. The plants are mainly Magnoliopsida | European Environment Agency (2019) [25] |

the Magnoliopsida and Lilopsida, though there are only 25 plants in the whole list. The animals included are from a wide variety of classes and orders including Insecta, Mollusca, Actinopterygii, Aves, Mammalia and Ascidiacea.

Also relevant is the Invasive Alien Species Regulation of the European Union (Regulation (EU) 1143/2014) (Table 1) which features a nearly equal representation of plants (41 species) and animals (47 species). The animals are mostly vertebrates, such as fish, birds, and mammals, and the plants are mostly Tracheophyta.

There are also a total of 9237 non-native species recorded in the European Union [41,42].

**3.1.4 Species listed in European Union directives.** The species listed in the Birds Directive, Habitats Directive and Marine Strategy Framework Directive largely focus on vertebrates, such as mammals, fish and birds, though the Habitats Directive does list some fungi and vascular plants (Table 1). The main taxonomic focus of the Pollinators Initiative are Bees (Anthophila) and hoverflies (Syrphidae) and butterflies (Papilionoidea).

## 3.2 The supply of taxonomic expertise

**3.2.1 Workflow results.** Of the 2,686 journals identified, we focused on the 1,103 journals with OpenAlex IDs (Table 2). However, only 474 of these journals were represented in the selected articles of European affiliated authors. This discrepancy arises because approximately 24% of the journals identified had dissolved before our period of interest, while many others have ceased publication but lack dissolution records in Wikidata. In a sample of 50 journal titles with no OpenAlex ID and no dissolved date, 41 were in fact no longer publishing, and others were unrelated to the biogeography of Europe, such as *Sansevieria* and the *University of Wyoming Publications in Science. Botany*. Only one journal from this sample had articles which could have been included if it had an OpenAlex ID; this was the *Atti della Società Toscana di Scienze Naturali, Memorie, Serie B* (Table 2). Most (92%) journals were uniquely identified through their IPNI or Zoobank ID. OpenAlex concepts (4.2%) and Wikidata subjects (1.0%) identified a small number of additional journals uniquely.

Searching these journals in OpenAlex resulted in 33,499 articles with at least one author with at least one European affiliation. A word cloud (S1 Fig) visualises the most common words occurring in the title and abstract of the articles found. In a random sample of 200 articles, we only found four articles that were not considered taxonomic in scope or 2%.

We identified 31,839 European authors, each with a unique OpenAlex ID, associated with taxonomic articles. Following disambiguation, this number was reduced to 31,521. A saturation curve of authors from those journals showed that further addition of journals was unlikely to significantly increase the number of publishing taxonomists we could identify (S2 Fig). A sample of 200 authors, all with the same truncated name format (first initial and last name), was reviewed. Among

**Table 2. A summary of the taxonomic journal selection results. Journals related to taxonomy and their relation to other journal identifiers. Only active journals with an OpenAlex ID were used to select taxonomic articles.**

| Category | Number |
|---|---|
| Journals identified and deduplicated | 2686 |
| Journals with a unique Wikidata ID | 2502 |
| Journals with a unique ISSN-L | 1791 |
| Journals with an OpenAlex ID | 1021 |
| Dissolved journals | 801 |
| Journals with no OpenAlex ID and not dissolved | 914 |
| Active journals that contained relevant articles | 474 |

these, 21 authors were merged by the system, and all mergers were deemed appropriate upon manual verification. However, nine pairs of names (4.5%) were not merged by the system, despite manual checks confirming they referred to the same individual. Authors with the same truncated names accounted for only 10% of all authors in the dataset. Consequently, the estimated total number of authors inadvertently included was less than 0.5%. While this may lead to a slight overestimation in the number of taxonomists, we deemed this acceptable within the scope of the analysis.

**3.2.2 Taxonomic expertise.** Where an author was able to be linked to a specific kingdom, i.e., in 57% of the cases, 59.3% published on Animalia, 38.0% on Plantae and 12.5% on Fungi. Seventeen percent had published on more than one kingdom. The ten most frequently studied families, based on the number of published articles, were Asteraceae (400 articles), Staphylinidae (311), Fabaceae (297), Orchidaceae (297), Poaceae (262), Scarabaeidae (173), Curculionidae (168), Lamiaceae (150), Erebidae (145), and Caryophyllaceae (139) (S3 Fig).

Plant taxonomists focus largely on vascular plants, largely the Magnolopsida, Liliopsida and the Pinopsida (Fig 2). Animal taxonomists have perhaps more diverse interests, but the number of active publishing taxonomists is concentrated on vertebrates and insects (Fig 2). However, this pattern should be interpreted in light of the extremely uneven distribution of species richness across animal groups. Insects comprise the majority of described animal species, so order-level counts of publishing taxonomists do not necessarily reflect proportional research coverage per species. For example, normalising by described species richness suggests that Phasmida has approximately three times more taxonomists per species than Coleoptera, and that among hyperdiverse orders Hymenoptera (0.059) has a higher taxonomist-per-species ratio than Hemiptera (0.035), Psocodea (0.030), Diptera (0.038) and Lepidoptera (0.038). Fungal taxonomists are dispersed across the kingdom, but not evenly. In the Basidiomycota the Agaricomycetes are most studied. Taxonomists are widely distributed across the Ascomycota, some in lichenizing groups such as the Lecanoromycetes, and Arthoniomycetes, other classes include numerous plant pathogens, such as Dothideomycetes, Sordariomycetes and Eurotiomycetes.

**3.2.3 Geographic distribution.** In Europe there is a clear east-west divide in the number of taxonomists (Fig 3A). However, as the number of taxonomists is strongly correlated with the population size of the country (Pearson r = 0.8640; p < 0.001; S4 Fig) on a per capita basis this division is weaker (Fig 3B). On a per capita basis, Iceland, Estonia, Norway and the Czech Republic, Switzerland and Portugal stand out as having a high proportion of taxonomists. While the number of taxonomists in a country will depend on many factors, we do see a strong correlation with the population of that country. As the collecting of voucher specimens is also an important aspect of professional taxonomy we also examined the number of unique collector and identifier strings and their distribution in Europe. Unlike article authors it is not possible to determine the affiliation of collectors, only where the specimens were collected. Nevertheless, a similar pattern to author affiliation was found, with eastern countries having apparently fewer collectors and identifiers identified (S5 Fig).

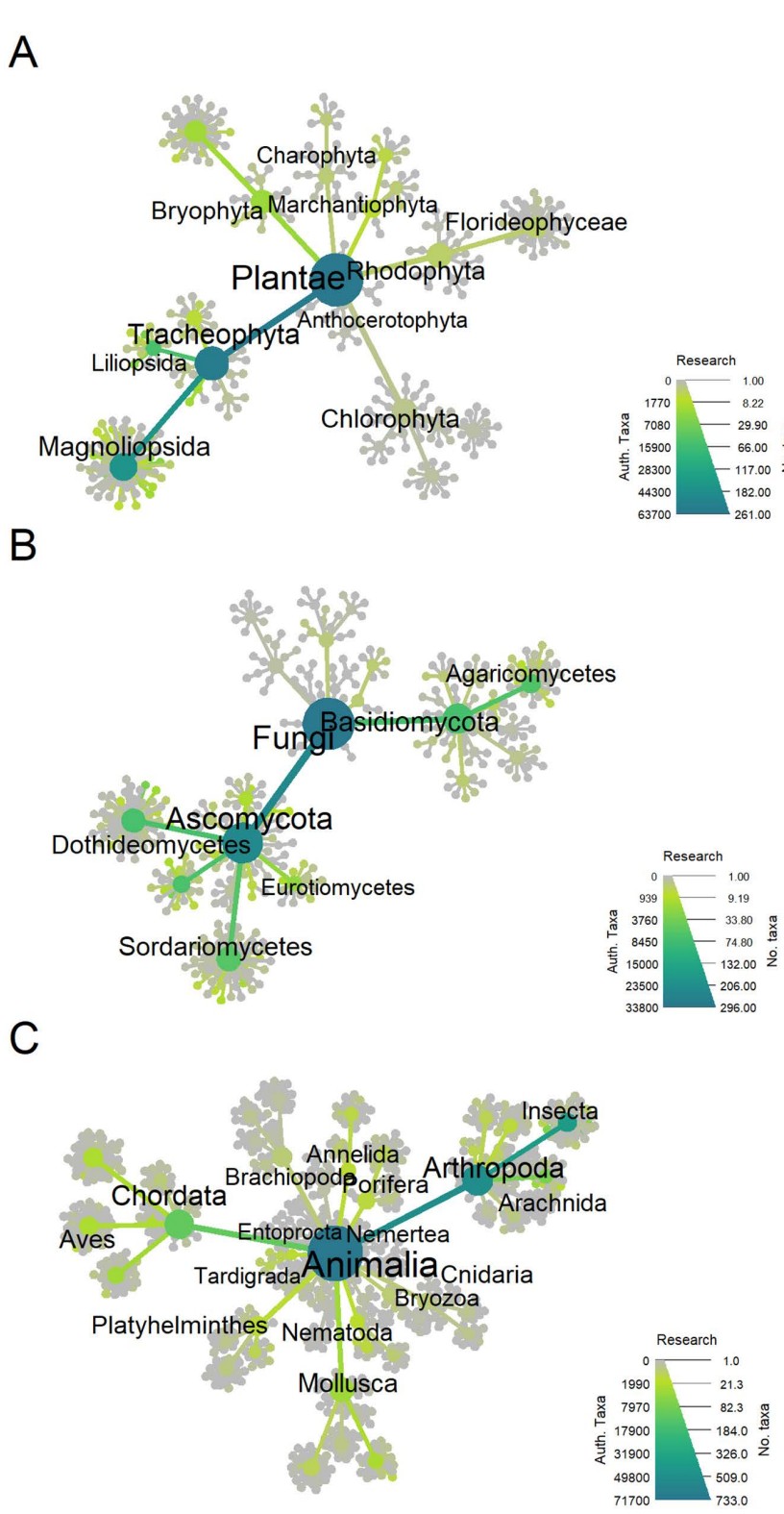

**Fig 2. Phylogenies of the publication activity of taxonomists separated into the kingdoms (A) Plantae (B) Fungi and (C) Animalia.** Tips represent orders, but only some classes and higher taxa are labeled.

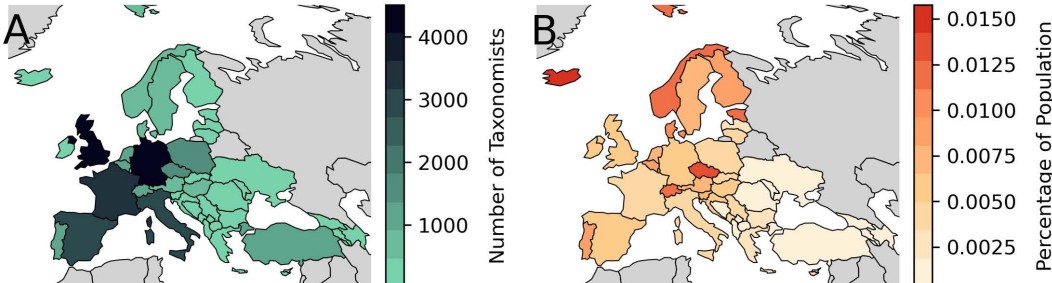

**Fig 3. The number of taxonomists affiliated with institutions within European countries as a (A) total and as a (B) percentage of the population of each country.** Free vector and raster map data @ naturalearthdata.com.

### 3.3 The relation between the number of named taxa, policies and taxonomists

The robust regression analysis revealed significant associations between species richness, biodiversity policies, and the number of authors involved in taxonomic research (S6 Fig). In the species-only model, species richness alone demonstrated a highly significant positive relationship with the number of authors ($\beta$ = 0.662, p<0.001). However, when biodiversity policy variables were included in the combined model, the strength of this relationship decreased ($\beta$ = 0.471, p<0.001), indicating that policy variables explained additional variance previously attributed solely to species richness.

Several policy-demand variables were significantly associated with taxonomic research effort, included explicit taxonomic research needs ($\beta$ = 0.541, p<0.001), the Habitats Directive ($\beta$ = 0.348, p=0.003), and the Birds Directive ($\beta$ = 0.369, p=0.041). Marine-related policies exhibited a negative but significant effect ($\beta$ = −0.273, p=0.038). Other policy variables, such as crop wild relatives, the IAS list of Concern, invasive species on the horizon, Red Listed species, and pollinators, did not reach statistical significance individually (p>0.05) (S7 Fig).

An F-test comparing the combined model with the species-only model confirmed that biodiversity policies significantly improved the model's explanatory power (p<0.05), underscoring the collective importance of policy factors beyond species richness alone.

Residual diagnostics indicated deviation from normality (Shapiro-Wilk test, p<0.001). While this suggests a violation of the normality assumption, robust regression methods such as those used in this analysis (RLM with Huber's T norm) are designed to mitigate the impact of non-normal residuals. Consequently, the deviation from normality observed here is unlikely to substantially affect the validity of our conclusions. Nevertheless, the examination of top residual outliers highlighted taxa potentially influenced by additional unmeasured factors, suggesting avenues for future investigation. The Durbin-Watson test result (1.975) confirmed that there was no significant autocorrelation among residuals, indicating independence of observations.

However, the Breusch-Pagan test for heteroscedasticity was highly significant (test statistic=309.45, p<0.001), suggesting that residual variance is not constant across predicted values. This finding indicates that some variables may exert disproportionate effects across different taxonomic groups. While robust regression reduces sensitivity to heteroscedasticity, this remains a limitation of the model. Further exploration of transformations or stratified analyses may be required in future research.

Despite these limitations, the overall model results provide strong evidence that biodiversity policy variables are associated with taxonomic research effort beyond what can be explained by species richness alone.

## 4 Discussion

The IUCN Red List, a cornerstone of conservation policy, exemplifies the critical need for taxonomic precision. It identifies a substantial proportion of European species requiring further taxonomic study, particularly among plants such as

Tracheophyta and animals, including Gastropoda and Insecta. Taxonomic assessments of these threatened species are essential for achieving targets like the EU Biodiversity Strategy's goal to reduce the impact of invasive alien species on vulnerable taxa. However, significant gaps in taxonomic expertise remain for fungi, algae, non-insect invertebrates, and for many insect groups beyond those currently covered by assessments.

While policies such as the EU Soil Strategy for 2030, Marine Strategy Framework Directive, and Habitats Directive may contribute to endangered species conservation, they fall short of fostering a deeper understanding of their taxonomy and conservation status. There is a catch-22 whereby rare species are not assessed because there is no one to study them, and there is no one to study them because there is no conservation policy driver until they are assessed. A positive example that broke this cycle was a specific focus on Bryophytes in the *Research Needed category* in European red-lists which is likely the result of a targeted project from a large group of dedicated bryologists, funded at least in part through the LIFE project of the European Commission, not necessarily *a prior* conservation priorities [43] A clear case of a known knowledge gap being addressed by funding and potentially leading to improved conservation outcomes. This provides a clear example of how limited expertise can constrain assessment, which constrains policy recognition, which in turn reduces incentives for expertise to develop.

This suggests that the relationship between policy and taxonomic activity is best understood as bi-directional. Policy incentives can stimulate research and assessment, but the availability of taxonomic expertise and information can also determine which taxa become visible to policy in the first place.

Our use of European Red List variables reflects explicit policy-facing conservation assessment efforts, but these assessments cover only selected taxonomic groups. As a result, many taxa, including numerous insect orders and most invertebrate phyla, are absent not because they face no extinction risk, but because they remain unassessed at the European level. For example, the European Red List includes a dedicated assessment for hoverflies (Diptera: Syrphidae), but not Diptera as a whole, and other ecologically specialised groups (e.g., parasitic lice or Collembola) are not currently assessed.

For invasive alien species, the demands are equally pressing. Horizon scans and the EU Invasive Alien Species Regulation prioritize taxonomic clarity for a diverse range of taxa, particularly vertebrates and vascular plants. With over 9,000 non-native species recorded in Europe, accurate identification is essential for effective management [41]. Similarly, crop wild relatives demand taxonomic attention, especially within economically significant families. These taxa underpin agricultural resilience and food security, highlighting the necessity of robust taxonomic frameworks to guide conservation [22]. Our findings complement earlier work that explicitly assessed taxonomic support needs for IAS management and the operational capacity of institutions to deliver rapid identifications in surveillance and interception contexts [26,27].

Legal instruments like the EU Birds Directive, Habitats Directive, and Marine Strategy Framework Directive predominantly focus on vertebrates but extend to select fungi and vascular plants. Taxonomic expertise is essential for compliance, monitoring, and biodiversity protection [44]. Additionally, the EU Pollinators Initiative, while not a directive, underscores the critical role of taxonomy in supporting pollinator conservation efforts, focusing on groups such as bees, hoverflies, and butterflies.

Our analysis uses scientific publications as a scalable proxy for taxonomic capacity, but we recognise that many of the most policy-relevant contributions of taxonomists are not captured by publication metadata alone. Moreover, publication activity over a limited time window reflects recent publishing effort and may underestimate longer-term capacity held by experienced specialists whose contributions are expressed primarily through identifications, curation, and synthesis rather than frequent indexed publication. These include red-list assessments and reassessments, identification tools and keys, curated checklists and nomenclatural updates used in reporting, expert identifications supporting monitoring schemes, and the development of reference collections and digitised specimen data that underpin biodiversity indicators. This limitation is particularly relevant for hyperdiverse groups such as insects, where, in Europe, substantial expertise is held by independent or amateur specialists and some recognised taxonomic authorities have no formal institutional affiliation.

This distinction is important because policy relevance is often expressed through such applied and infrastructure outputs, which are not consistently indexed, standardised, or linked through persistent identifiers across EU portals. Expanding this framework to include metrics such as expert identifications, checklist contributions, collection curation and digitisation, and advisory participation would provide a more complete picture of how taxonomic effort supports policy implementation.

Our publication-based indicators capture taxonomic research activity and expertise located in Europe, but do not systematically distinguish whether the taxa are European or non-European. Consequently, some Europe-based taxonomists publishing primarily on non-European taxa are included despite limited direct relevance to European policy needs; resolving this at scale would require robust links between publications and study localities or species ranges, which are not consistently available, particularly in a machine readable format. This methodological limitation also reflects a broader strategic question of how European taxonomic capacity should be balanced between domestic policy needs and global biodiversity responsibilities.

Therefore overall in the policy landscape, groups including vertebrates, vascular plants, and some insect groups receive substantial attention. The question then is, are these policy demands met by the supply of taxonomists, and if so is it the policies that drive the demand for taxonomy, or the taxonomists that lead the formulation of policy? In practice, both processes likely operate simultaneously, consistent with co-production dynamics.

Our automated, reproducible workflow demonstrates the potential for large-scale systematic assessments of taxonomic capacity or the "supply of taxonomists" using open bibliographic resources. This kind of capacity assessment has been identified by the Convention on Biological Diversity (CBD) and its Global Taxonomy Initiative (GTI) as a critical foundation for effective biodiversity policy implementation [45]. The workflow supports not only global analyses but can also be adapted to address the specific challenges and priorities of particular regions or taxonomic groups, offering tailored metrics for national and institutional reporting. As compared to manual methodologies for inventorizing taxonomic capacity, often limited in taxonomic and/or geographical scope (e.g., [23,46,47], standardised methodologies like this ensure that trends in authorship, focus, and capacity are monitored over time, providing a robust evidence base for decision-making in biodiversity conservation, resource allocation, and strategic planning.

Our workflow identifies a large number of taxonomic journals, articles, and their associated authors. OpenAlex, the bibliographic resource we rely on, claims to have about twice the coverage of comparable services, including better representation of non-English works (https://docs.openalex.org/). While we cannot claim to achieve complete coverage—especially given the minimal restrictions on where taxonomic acts can be published—our focus is on mainstream, peer-reviewed journals where most professional taxonomists in Europe publish. Also, while errors do exist in the used open resources—just because they are open—these errors are visible, and in the case of Wikidata, directly correctable by users. OpenAlex and the Taxonomic Backbone are not directly editable but are dynamic and incorporate user feedback.

The results of our workflow reveal an uneven geographic distribution of taxonomic expertise in Europe, with northern and western regions hosting the majority of taxonomists. Even on a per capita basis, eastern European countries have fewer taxonomists, though countries like Estonia, the Czech Republic, and Portugal stand out for their high concentration of taxonomists relative to their population.

Among plant taxonomists the focus is on vascular plants, especially large and economically significant families. Conversely, groups like algae (particularly microalgae) remain understudied despite their taxonomic diversity. Among animals, vertebrates and insects receive the most attention, whereas aquatic non-insect invertebrates are noticeably underserved. For example, there are few taxonomists for phyla as Annelida, Brachiopoda, Cnidaria and Porifera. Many fungal groups are particularly underrepresented despite their ecological and economic importance. For example, fungal taxonomy is concentrated in a few classes like Agaricomycetes (Basidiomycota) and some plant-pathogenic Ascomycota. Taxonomy operates in a publicly funded, non-market-driven system where supply and demand are not tightly coupled. Instead, taxonomic capacity is determined by institutional priorities and funding availability, often influenced by decision makers' understanding of societal needs. This potential misalignment allows for both oversupply and unmet demand within the field, contributing to the "taxonomic impediment".

 

While our study focuses on scientific publication as a measure of taxonomic capacity and examines only a sample of demand-side drivers, it highlights critical trends. Expanding this evaluation to include other metrics, such as teaching, specimen collection, curation, and digitization, could provide a more comprehensive picture of taxonomic capacity and its alignment with societal demands.

Our results do demonstrate that there is some relationship between the number of taxonomists and the attention demanded by policy. For example, vertebrates are frequently mentioned in EU policy, and indeed there are a comparatively high number of vertebrate taxonomists. Likewise, there are many taxonomists of vascular plants and specific policy drivers linked to these. However, a close correlation between human population in a country and the number of taxonomists is probably because taxonomists in populous countries are often working on taxa outside their affiliated country, rather than populous countries being more biodiverse, or that policies in populous countries demand more taxonomists [48,49].

Furthermore, it is not always possible to determine logical reasons for research funding decisions, because they are often influenced by subjective factors, such as the perceived charisma or public appeal of certain species, rather than being strictly guided by clear, data-driven conservation priorities [50]. Additional drivers likely include the distribution of historical collections and infrastructure, publication incentives, and the differential visibility of taxa to the public and to monitoring schemes. These drivers are not mutually exclusive and may reinforce one another, producing persistent mismatches between research attention and policy needs. Our methodology does not directly distinguish policy-driven taxonomic research from indirect research on policy relevant species. A fuller causal assessment would require integrating funding, institutional capacity, and expert-infrastructure indicators, which we suggest as a direction for future work.

## 4.1 Recommendations for taxonomists, their institutions and their funders

The uneven geographic and taxonomic distribution of expertise underscores the need for better coordination among funders, institutions, researchers, policymakers, and those responsible for policy implementation, including environmental agencies, monitoring coordinators, and regulatory authorities. Systematic approaches to measure supply and demand, coupled with strategic partnerships, can help align taxonomic efforts with societal priorities, ensuring that outputs are impactful, adequately funded, and actionable for biodiversity conservation and sustainable management [51].

Effective collaboration between taxonomists, policymakers, conservation practitioners, and funders requires structured mechanisms for engagement. Platforms such as multistakeholder forums, regional or international workshops, and decentralized networks can facilitate co-developed research agendas, shared learning, and inclusive decision-making [52,53].

To further enhance coordination, national or regional taxonomic advisory councils—comprising representatives from government, academia, NGOs, and industry—could help identify priorities, guide funding, and target underrepresented taxa and regions (e.g., aquatic invertebrates, fungi; [54]. Joint funding initiatives among public and private entities can also amplify impact by supporting research in biodiversity-rich areas or on neglected groups. Nevertheless, coordination alone is unlikely to be effective unless the incentives shaping research investment and career progression are also aligned with policy needs, since funding priorities, evaluation metrics, and publication incentives do not always reward taxonomic synthesis and applied outputs, even when these are needed for policy implementation

Embedding taxonomic expertise within policymaking bodies would strengthen the science-policy interface. Seconding taxonomists to environmental agencies, for instance, ensures that policy decisions are grounded in up-to-date taxonomic knowledge [55]. At the same time, metrics and reporting systems can help track how research contributes to policy objectives, conservation outcomes, or global biodiversity frameworks [56,57].

Building capacity remains essential in bridging the gap between the supply and demand for taxonomic expertise. Investments in training, research infrastructure, and collections that underpin biodiversity studies, will ensure that the field

has the resources it needs to thrive. Training should include communication and policy engagement skills, while policy-makers could receive training on the significance of taxonomy and its application in decision-making, fostering mutual understanding and enhancing collaboration. Recognizing policy-engaged academic work in career advancement would further incentivize impactful research.

More specifically, we recommend:

1. Encourage taxonomic publications, checklists, and identification tools to include persistent taxonomic identifiers (e.g., GBIF, Catalogue of Life, EUNIS IDs) and policy tags (e.g., Birds/Habitats Directive annex species; IAS Regulation; MSFD indicator taxa) to support automated linkage between taxonomy and policy reporting;

2. Fund revisions and checklists explicitly aligned with EU reporting cycles (e.g., Article 17, Marine Strategy Framework Directive assessments), prioritising groups where our residual analysis indicates undersupply;

3. Maintain and modernise collections, barcoding and imaging capacity, and digitisation pipelines as essential foundations for both taxonomic research and applied policy outputs;

4. Formalise secondments and joint appointments between research institutions, policymakers and policy-implementing agencies to accelerate translation of taxonomic advances into regulatory and management decisions.

Together, these measures can create an integrated and responsive taxonomic system. Co-developing goals ensures that research priorities reflect both scientific and policy needs, while shared evaluation frameworks and feedback mechanisms maintain accountability and mutual trust. Ultimately, shared ownership of outcomes—whether in improved policies or research advances—will sustain and strengthen partnerships between science and society.

Lastly, a substantial component of European taxonomic expertise has historically been directed toward biodiversity outside Europe, including tropical regions, and this remains important for global biodiversity knowledge and for supporting countries with limited taxonomic capacity. However, it is often unclear how this extraterritorial role is valued and prioritised relative to domestic biodiversity policy needs, particularly in light of evolving expectations under the CBD and access-and-benefit-sharing. We recommend that European funders and policymakers explicitly articulate strategic priorities for taxonomic research both within and beyond Europe, and develop equitable partnership and capacity-building models to ensure that externally focused taxonomy delivers mutual benefits while maintaining sufficient capacity to meet European policy obligations.

Looking ahead, taxonomy stands at a transformative juncture. Innovations like artificial intelligence, metabarcoding, and automated monitoring tools offer vast new data and insight. While this data deluge presents challenges, it also empowers taxonomists to redefine their role and deepen their societal impact. Harnessing these technologies will require sustained collaboration across disciplines and sectors. By seizing this moment of change, the taxonomic community can ensure that its work remains not only scientifically rigorous, but also essential to the protection of biodiversity and the well-being of future generations.

## 5 Conclusion

Taxonomy is indispensable for addressing biodiversity challenges, yet our results indicate that publication-based taxonomic research effort is only partially aligned with major biodiversity policy demands, and substantial gaps persist for several policy-relevant groups. By adopting systematic approaches to assess and monitor taxonomic capacity, we can better prioritize investments and guide taxonomists toward impactful research. The automated, reproducible workflow developed in this study serves as a foundation for future work, enabling more informed decisions that support biodiversity conservation and sustainable development.

## Supporting information

**S1 Fig. A word cloud to visualise the most common words occurring in title and abstract of the 33,499 articles found.** This gives a visual representation of the subjects of the articles and a qualitative check that we have predominantly filtered for taxonomic articles.
(TIF)

**S2 Fig. The cumulative frequency curve of newly discovered authors using 474 journals after author deduplication (see methods).**
(TIF)

**S3 Fig. A histogram showing the familial focus of taxonomic articles with authors affiliated to European institutions.** The top 10 families by number of articles are Asteraceae: 400, Staphylinidae: 311, Fabaceae: 297, Orchidaceae: 297, Poaceae: 262, Scarabaeidae: 173, Curculionidae: 168, Lamiaceae: 150, Erebidae: 145, Caryophyllaceae: 139.
(TIF)

**S4 Fig. Correlation of population size and number of taxonomists per country (N = 48) in Europe.** Pearson correlation coefficient: 0.8640 (p = 2.634e-15).
(TIF)

**S5 Fig. The distribution of distinct dwc:recordedby text strings.** (A) and dwc:identifiedBy text strings (B) on occurrences from GBIF per country from years between 2014 and 2023 inclusive. Note that the colour ramp is on a logarithmic scale and that the maps use an equal area Mollweide projection. Made with Natural Earth. Free vector and raster map data @ naturalearthdata.com.
(TIF)

**S6 Fig. Summary diagnostics and regression results for the log–log robust regression model.** (A) Estimated regression coefficients (±95% confidence intervals) for the model relating taxonomic research effort (number of authors) to species richness and policy variables. Predictors are ordered by absolute importance. (B) Relative importance of predictors based on the absolute value of their standardised regression coefficients. (C) Standardised residuals from the robust regression model, with horizontal red lines indicating thresholds for potential outliers (residual > 3). (D) Q–Q plot showing the quantiles of the residuals against the theoretical quantiles of a normal distribution. Deviation from the 45° reference line indicates departures from normality.
(TIF)

**S7 Fig. Scatterplots of policies and plant taxonomists showing the correlation between the policy relevance of each plant order and the number of taxonomists working on them.** Each plot corresponds to a different policy included in this study.
(TIF)

**S8 Fig. Scatterplots of policies and animal taxonomists showing the correlation between the policy relevance of each animal order and the number of taxonomists working on them.** Each plot corresponds to a different policy included in this study.
(TIF)

**S9 Fig. Scatterplots of policies and fungal taxonomists showing the correlation between the policy relevance of each fungal order and the number of taxonomists working on them.** Each plot corresponds to a different policy included in this study.
(TIF)

**S1 Table. The most common taxa in the European Red List, the European Red List taxonomic research needed category, the list of European crop wild relatives, and invasive species on a horizon scanning list.** Numbers in parentheses indicate the number of species in each taxon.
(PDF)

## Acknowledgments

We thank the reviewers for their constructive and insightful comments. This work would not have been possible without the open data infrastructures provided by OpenAlex, Wikidata, and GBIF, and the efforts of their respective communities. We also acknowledge the TETTRIs and B3 project consortia and the wider DiSSCo community for shaping discussions on taxonomic capacity and its role in biodiversity policy.

## Author contributions

**Conceptualization:** Quentin J. Groom, Sofie Meeus.

**Data curation:** Lina M. Estupinan-Suarez.

**Formal analysis:** Quentin J. Groom.

**Funding acquisition:** Quentin J. Groom.

**Investigation:** Quentin J. Groom.

**Methodology:** Quentin J. Groom.

**Project administration:** Quentin J. Groom, Sofie Meeus.

**Software:** Quentin J. Groom, Melanie De Nolf.

**Supervision:** Sofie Meeus.

**Validation:** Quentin J. Groom.

**Writing – original draft:** Quentin J. Groom.

**Writing – review & editing:** Quentin J. Groom, Lina M. Estupinan-Suarez, Sofie Meeus.

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
