## [Decision Letter · Decision Letter 0]

18 Nov 2025

Dear Dr.  Groom,

Thank you for submitting your manuscript to PLOS ONE. After careful consideration, we feel that it has merit but does not fully meet PLOS ONE’s publication criteria as it currently stands. Therefore, we invite you to submit a revised version of the manuscript that addresses the points raised during the review process.

We look forward to receiving your revised manuscript.

Kind regards,

Daniel de Paiva Silva, Ph.D.

Academic Editor

PLOS ONE

Journal Requirements:

2. We note that Figure 3 and Figure S5 in your submission contain map images which may be copyrighted. All PLOS content is published under the Creative Commons Attribution License (CC BY 4.0), which means that the manuscript, images, and Supporting Information files will be freely available online, and any third party is permitted to access, download, copy, distribute, and use these materials in any way, even commercially, with proper attribution. For these reasons, we cannot publish previously copyrighted maps or satellite images created using proprietary data, such as Google software (Google Maps, Street View, and Earth). For more information, see our copyright guidelines: http://journals.plos.org/plosone/s/licenses-and-copyright.

1. You may seek permission from the original copyright holder of Figure 3 and Figure S5 to publish the content specifically under the CC BY 4.0 license.

3. Please include captions for your Supporting Information files at the end of your manuscript, and update any in-text citations to match accordingly. Please see our Supporting Information guidelines for more information: http://journals.plos.org/plosone/s/supporting-information....

Additional Editor Comments:

Dear Dr. Groom,

After this first review round, both reviewers were very positive regarding your manuscript. Therefore, they indicated the need for minor reviews (or even the "Acceptance" status). Therefore, as soon as you insert the improvements suggested by one of the reviewers, and once the reviewer agrees the changes were satifatory, I am sure your manuscript will be accepted for publication in PLoS One. Congratulations on your hard work.

Sincerely,

Daniel Silva

Reviewer's Responses to Questions

**Comments to the Author**

1. Is the manuscript technically sound, and do the data support the conclusions?

Reviewer #1: Yes

Reviewer #2: Yes

2. Has the statistical analysis been performed appropriately and rigorously?

Reviewer #1: Yes

Reviewer #2: Yes

3. Have the authors made all data underlying the findings in their manuscript fully available?

Reviewer #1: Yes

Reviewer #2: Yes

4. Is the manuscript presented in an intelligible fashion and written in standard English?

Reviewer #1: Yes

Reviewer #2: Yes

Reviewer #1: Groom et al. present a timely and very well-written manuscript aimed at quantifying taxonomic output by European-based researchers and the level of demand for specific taxonomic knowledge. I found no real issues with the manuscript, and would be quite happy to see it proceed for publication.

One or two things which could be noted and/or briefly discussed are publication activity does not necessarily equal taxonomic capacity, particularly within a 10 year time frame. This ties to my second point which is the knowledge held within the amateur community. In my group (bees), but more broadly found in the insect world, many taxonomic authorities do not have institutional affiliations. If I consider the people working on the EPIC (European Pollinator Identification Courses) project (tied to the EU-PoMS monitoring scheme that you mention in the manuscript), perhaps 70% of the taxonomists have an institutional affiliation. This is actually higher than is representative for bees more broadly (there are some experts who were just not interested in participating). Whilst I fully understand your decision to make a transparent reproducible workflow based on accessible databases, acknowledging that this will not capture absolutely all the existing taxonomic knowledge base could be beneficial.

I would also highlight that there is a slight tension in the methodology and the goal of matching knowledge:demand in a European context. Your metric for selecting publications does not (from what I can see) distinguish between research on European taxa, and non-European taxa. A botanist working here at Naturalis publishing on West African plants will appear as "active" in your workflow, but does not actually provide appropriate knowledge to meet EU policy objectives focused on Europe. Now, distinguishing between European and non-European taxa in your workflow could be a big headache, and I understand why you did not do it. Many taxonomists working on non-European taxa also have good knowledge of their local fauna/flora. So, this issue may be academic, but I think that it is worth acknowledging.

With my best wishes,

Thomas Wood

Reviewer #2: 1. This is an important topic and the authors provide an interesting analysis. It would perhaps have been valuable to investigate drivers of the current situation; the existence of the taxonomic impediment to policy is not new – it has been discussed numerous times. Quantification, albeit at a high level, is helpful, although the recommendations at the end of the paper could be more extensive. Although the abstract states “We explore how this supply of expertise compares with the kinds of demands that arise from European biodiversity policy, including legally binding instruments such as the Birds and Habitats Directives and the Marine Strategy Framework Directive, as well as strategic initiatives focused on invasive alien species, crop wild relatives, and species of conservation concern”, this is not a strong element of the paper, which focusses more on availability of expertise rather than what those experts are producing and whether or not it is policy-relevant.

2. While the term ‘taxonomic impediment’ is stated in the paper to be “the challenge of cataloguing Earth’s biodiversity” this is a limited characterisation, missing the purpose of the original coining of the term. The term is better used to identify the impediment to implementing policy objectives due to the lack of appropriate taxonomic expertise and information. It has been used in this way by the CBD and, indeed, is the topic of the paper. A clear statement early in the paper would be helpful to set the stage.

3. When referring to Invasive Alien Species the authors might like to refer to a couple of relevant works that focussed on the relevant taxonomic needs: a) Lyal, C.H.C. & Miller, S.E., 2019, Capacity of United States federal government and its partners to rapidly and accurately report the identity (taxonomy) of non-native organisms intercepted in early detection programs. Biological Invasions https://doi.org/10.1007/s10530-019-02147-x 27pp. b) Smith, R.D., Aradottir, G.I., Taylor, A. & Lyal, C., 2008, Invasive species management – what taxonomic support is needed? Global Invasive Species Programme, Nairobi, Kenya. 52pp.

4. P. 9. When referring to the European Red list there seems to be no analysis of the ‘missing’ taxa, other than a statement that “classes like Clitellata, Echinodermata, and Platyhelminthes are apparently underrepresented”. This applies also to many of the insect groups, for example, and a comparison of the groups represented in the red list compared to, e.g. field guides, might be informative as to the relevance of accessible taxonomic information to the compilation of the Red Lists. A brief inspection, for examples, shows that the only Diptera included are in the Syrphidae – it seems implausible that no other Diptera are threatened. There are no Phthiraptera, although their host specificity suggests that for every endangered bird or mammal there are likely to be one or more endangered louse species. There are no Collembola – is this due to their conservation security or a lack of knowledge of the group as a whole? I suggest that the point on p. 12 “However, significant gaps in taxonomic expertise remain for fungi, algae, and non-insect invertebrates” should acknowledge gaps in other groups (including insects) where no work has been done at all.

5. P. 11. “Animal taxonomists have perhaps more diverse interests, but are mainly focused on vertebrates and insects.” This is doubtless correct, but the sizes of the various groups could be taken into account. Within the insects there are vastly more species than in the other groups considered, and I wonder if a more appropriate level here may be Order. The listing of families studied suggests an uneven degree of taxonomic attention, and it would be illustrative if the ‘bottom 10’, or a list of families with no publications, could also be listed.

6. P. 11. A causative relationship is inferred for the correlation between taxonomic research and biodiversity policy (“Specific biodiversity policies independently influencing taxonomic research…”). This may be so, but it has already been implied that the lack of taxonomic information can influence policy coverage and prioritisation, leading to a potentially circular argument. One can view policies as to an extent co-productions of policymakers and experts; if there is no available taxonomic expertise in a group is may not be recognised by policymakers at all, since it has no expert advocacy. Higher levels of taxonomic information and expertise enables inclusion of taxa covered in policies, and consequently delivers a correlation between policy coverage and research activity. The final sentence of section 3.3. “Despite these limitations, the overall model results provide strong evidence that biodiversity policy is related to taxonomic research effort, beyond what can be explained by species richness alone” in stressing correlation not causation is more appropriate. This is captured on P. 12 by “There is a catch-22 whereby rare species are not assessed because there is no one to study them, and there is no one to study them because there is no conservation policy driver until they are assessed.” Absolutely! The bryological example underlines this potential and sometimes actual circularity.

7. While the paper identifies a correlation between number of taxonomists and the attention demanded by policy (p.14), it could go further in considering a third factor, the charismatic nature of the group (the Red List containing more butterflies than moths, only hoverflies amongst the Diptera, a strong focus of research on vertebrates etc). This might drive both taxonomic attention and elements of policy focus, particularly in conservation priorities. In referring to taxonomic research, this could lead to the correlations reported. However, while there may be more research on such groups is the research policy focussed or simply policy adjacent.

8. The recommendations are very important, but could be expanded. I would note that the identified need for “better coordination among funders, institutions, and researchers” should also include policymakers and those responsible for policy implementation. Funding bodies are not always responsive to environmental policies, having in some cases very different drivers from policymakers. Without considering the aspects that influence what research is done (e.g. grant availability and grant success, citation index, impact factor of journals etc for scientific institutions and scientist career progression, requirements for cutting edge research for some science funders), coordination alone will not have a strong impact. It would also be helpful to understand more what policy needs, to provide a pointer to how discussions should be focussed.

.

Reviewer #1: **Yes:** Thomas James WoodThomas James WoodThomas James WoodThomas James Wood

Reviewer #2: No

---

## [Author Response · Author response to Decision Letter 1]

20 Feb 2026

Reviewer #1:

"Groom et al. present a timely and very well-written manuscript aimed at quantifying taxonomic output by European-based researchers and the level of demand for specific taxonomic knowledge. I found no real issues with the manuscript, and would be quite happy to see it proceed for publication.

One or two things which could be noted and/or briefly discussed are publication activity does not necessarily equal taxonomic capacity, particularly within a 10 year time frame. This ties to my second point which is the knowledge held within the amateur community. In my group (bees), but more broadly found in the insect world, many taxonomic authorities do not have institutional affiliations. If I consider the people working on the EPIC (European Pollinator Identification Courses) project (tied to the EU-PoMS monitoring scheme that you mention in the manuscript), perhaps 70% of the taxonomists have an institutional affiliation. This is actually higher than is representative for bees more broadly (there are some experts who were just not interested in participating). Whilst I fully understand your decision to make a transparent reproducible workflow based on accessible databases, acknowledging that this will not capture absolutely all the existing taxonomic knowledge base could be beneficial."

We have expanded the Discussion to clarify that publication-based metrics over a limited time window represent recent publishing activity rather than the full taxonomic knowledge base, which also includes long-standing expertise expressed through identifications, curation, synthesis and applied outputs. We also explicitly acknowledge that substantial expertise in Europe resides in independent and amateur communities and may not be captured in bibliographic databases or affiliation metadata.

"I would also highlight that there is a slight tension in the methodology and the goal of matching knowledge:demand in a European context. Your metric for selecting publications does not (from what I can see) distinguish between research on European taxa, and non-European taxa. A botanist working here at Naturalis publishing on West African plants will appear as "active" in your workflow, but does not actually provide appropriate knowledge to meet EU policy objectives focused on Europe. Now, distinguishing between European and non-European taxa in your workflow could be a big headache, and I understand why you did not do it. Many taxonomists working on non-European taxa also have good knowledge of their local fauna/flora. So, this issue may be academic, but I think that it is worth acknowledging."

We agree that our publication-based proxy does not distinguish between expertise applied to European versus non-European taxa. We have added a limitation statement acknowledging this, and we also include a recommendation that European policymakers and funders clarify how taxonomic work beyond Europe is valued and prioritised relative to domestic biodiversity policy needs, particularly in the context of access and benefit sharing expectations and equitable partnership models.

Reviewer #2:

"1. This is an important topic and the authors provide an interesting analysis. It would perhaps have been valuable to investigate drivers of the current situation; the existence of the taxonomic impediment to policy is not new – it has been discussed numerous times. Quantification, albeit at a high level, is helpful, although the recommendations at the end of the paper could be more extensive. Although the abstract states “We explore how this supply of expertise compares with the kinds of demands that arise from European biodiversity policy, including legally binding instruments such as the Birds and Habitats Directives and the Marine Strategy Framework Directive, as well as strategic initiatives focused on invasive alien species, crop wild relatives, and species of conservation concern”, this is not a strong element of the paper, which focusses more on availability of expertise rather than what those experts are producing and whether or not it is policy-relevant."

We agree that the taxonomic impediment itself is well established and has been discussed extensively. Our intention was not to re-argue its existence but to contribute an extensive, reproducible and scalable approach for quantifying capacity using open bibliographic resources and for testing, at a high level, whether the distribution of this capacity corresponds to key biodiversity policy demands.

To address the reviewer’s concern that the policy-demand component was not sufficiently prominent, we have revised the abstract, introduction and discussion to better reflect this element of the study. In particular, we now highlight the robust regression results showing that policy variables collectively improve explanatory power beyond species richness alone, and that the Birds and Habitats Directives are positively associated with taxonomic research effort while marine-related policy variables show a negative association.

Finally, we have expanded the recommendations section with additional actionable proposals to strengthen the science-policy interface and to support targeted investment in underrepresented but policy-relevant taxonomic groups.

"2. While the term ‘taxonomic impediment’ is stated in the paper to be “the challenge of cataloguing Earth’s biodiversity” this is a limited characterisation, missing the purpose of the original coining of the term. The term is better used to identify the impediment to implementing policy objectives due to the lack of appropriate taxonomic expertise and information. It has been used in this way by the CBD and, indeed, is the topic of the paper. A clear statement early in the paper would be helpful to set the stage.

We agree and have revised the opening paragraph to define the “taxonomic impediment” in the sense used by the Convention on Biological Diversity, as the constraints that insufficient taxonomic knowledge, infrastructure and expertise impose on the implementation of biodiversity policy and management objectives, rather than only the challenge of cataloguing biodiversity."

"3. When referring to Invasive Alien Species the authors might like to refer to a couple of relevant works that focussed on the relevant taxonomic needs: a) Lyal, C.H.C. & Miller, S.E., 2019, Capacity of United States federal government and its partners to rapidly and accurately report the identity (taxonomy) of non-native organisms intercepted in early detection programs. Biological Invasions https://doi.org/10.1007/s10530-019-02147-x 27pp. b) Smith, R.D., Aradottir, G.I., Taylor, A. & Lyal, C., 2008, Invasive species management – what taxonomic support is needed? Global Invasive Species Programme, Nairobi, Kenya. 52pp."

We have added the recommended references to the sections discussing invasive alien species policy demand and the taxonomic expertise required for early detection and rapid response. Thank you for pointing these out.

"4. P. 9. When referring to the European Red list there seems to be no analysis of the ‘missing’ taxa, other than a statement that “classes like Clitellata, Echinodermata, and Platyhelminthes are apparently underrepresented”. This applies also to many of the insect groups, for example, and a comparison of the groups represented in the red list compared to, e.g. field guides, might be informative as to the relevance of accessible taxonomic information to the compilation of the Red Lists. A brief inspection, for examples, shows that the only Diptera included are in the Syrphidae – it seems implausible that no other Diptera are threatened. There are no Phthiraptera, although their host specificity suggests that for every endangered bird or mammal there are likely to be one or more endangered louse species. There are no Collembola – is this due to their conservation security or a lack of knowledge of the group as a whole? I suggest that the point on p. 12 “However, significant gaps in taxonomic expertise remain for fungi, algae, and non-insect invertebrates” should acknowledge gaps in other groups (including insects) where no work has been done at all."

We agree that our original wording could be interpreted as suggesting that taxa absent from the European Red Lists are not threatened, whereas in many cases they are simply not assessed, reflecting uneven taxonomic coverage of the European Red List programme. We have therefore revised the Methods and Discussion to clarify that ‘missing taxa’ primarily indicate gaps in assessment coverage and to acknowledge that substantial gaps remain not only for fungi, algae and non-insect invertebrates, but also for many insect groups beyond those currently assessed.

"5. P. 11. “Animal taxonomists have perhaps more diverse interests, but are mainly focused on vertebrates and insects.” This is doubtless correct, but the sizes of the various groups could be taken into account. Within the insects there are vastly more species than in the other groups considered, and I wonder if a more appropriate level here may be Order. The listing of families studied suggests an uneven degree of taxonomic attention, and it would be illustrative if the ‘bottom 10’, or a list of families with no publications, could also be listed."

We agree that statements about the distribution of taxonomic expertise should be interpreted in the context of the highly uneven distribution of species richness across animal groups, particularly the dominance of insects. We have revised the text accordingly and clarified that our metric reflects the number of active publishing taxonomists rather than publication volume. To account for group size effects, we also inspected insect expertise at Order level by normalising the number of publishing taxonomists by described species richness. We now include illustrative examples in the manuscript (e.g., Phasmida having approximately three times more taxonomists per species than Coleoptera, and differences among hyperdiverse orders such as Hymenoptera, Diptera, Lepidoptera, Hemiptera and Psocodea). We did not add an additional table because insects represent only one component of the broader taxonomic scope of the study, but these examples demonstrate the key point that taxonomic attention is not simply proportional to species richness.

"6. P. 11. A causative relationship is inferred for the correlation between taxonomic research and biodiversity policy (“Specific biodiversity policies independently influencing taxonomic research…”). This may be so, but it has already been implied that the lack of taxonomic information can influence policy coverage and prioritisation, leading to a potentially circular argument. One can view policies as to an extent co-productions of policymakers and experts; if there is no available taxonomic expertise in a group is may not be recognised by policymakers at all, since it has no expert advocacy. Higher levels of taxonomic information and expertise enables inclusion of taxa covered in policies, and consequently delivers a correlation between policy coverage and research activity. The final sentence of section 3.3. “Despite these limitations, the overall model results provide strong evidence that biodiversity policy is related to taxonomic research effort, beyond what can be explained by species richness alone” in stressing correlation not causation is more appropriate. This is captured on P. 12 by “There is a catch-22 whereby rare species are not assessed because there is no one to study them, and there is no one to study them because there is no conservation policy driver until they are assessed.” Absolutely! The bryological example underlines this potential and sometimes actual circularity."

We have revised the results to avoid causal language and to emphasise that our regression analyses identify statistical associations, not causal effects. We now explicitly acknowledge the likely bi-directional and potentially circular relationship between taxonomic capacity and policy coverage, consistent with a co-production perspective in which policy priorities and scientific expertise shape one another. We have strengthened the discussion of this “catch-22” dynamic and use the bryological example.

"7. While the paper identifies a correlation between number of taxonomists and the attention demanded by policy (p.14), it could go further in considering a third factor, the charismatic nature of the group (the Red List containing more butterflies than moths, only hoverflies amongst the Diptera, a strong focus of research on vertebrates etc). This might drive both taxonomic attention and elements of policy focus, particularly in conservation priorities. In referring to taxonomic research, this could lead to the correlations reported. However, while there may be more research on such groups is the research policy focussed or simply policy adjacent."

We agree that socio-cultural drivers such as a taxon’s visibility or charisma may influence both research attention. We have added a brief statement to the discussion acknowledging this as a limitation and clarifying that our publication-based metrics do not distinguish policy-driven from policy-adjacent research. We also acknowledge that future research could take this into account.

"8. The recommendations are very important, but could be expanded. I would note that the identified need for “better coordination among funders, institutions, and researchers” should also include policymakers and those responsible for policy implementation. Funding bodies are not always responsive to environmental policies, having in some cases very different drivers from policymakers. Without considering the aspects that influence what research is done (e.g. grant availability and grant success, citation index, impact factor of journals etc for scientific institutions and scientist career progression, requirements for cutting edge research for some science funders), coordination alone will not have a strong impact. It would also be helpful to understand more what policy needs, to provide a pointer to how discussions should be focussed."

We agree with the reviewer that coordination must include policymakers and those responsible for implementation, and we have revised the recommendations to state this explicitly. We also now note that coordination alone is unlikely to be effective unless the incentive structures shaping research funding and academic career progression are aligned with policy needs, because funding and evaluation frameworks do not always reward taxonomic synthesis and applied outputs. Finally, we have added a concise summary of recurring policy needs from taxonomy (e.g., authoritative names/identifiers, checklists, and identification capacity underpinning monitoring, reporting and risk assessment) to help focus future science-policy discussions.

---

## [Decision Letter · Decision Letter 1]

1 Apr 2026

Balancing the supply and demand for taxonomy: An analysis of European taxonomic capacity and policy needs.

PONE-D-25-45522R1

Dear Dr. Groom,

We’re pleased to inform you that your manuscript has been judged scientifically suitable for publication and will be formally accepted for publication once it meets all outstanding technical requirements.

Kind regards,

Daniel de Paiva Silva, Ph.D.

Academic Editor

PLOS One

Additional Editor Comments (optional):

Reviewers' comments:

Reviewer's Responses to Questions

**Comments to the Author**

Reviewer #2: (No Response)

2. Is the manuscript technically sound, and do the data support the conclusions?

Reviewer #2: Yes

3. Has the statistical analysis been performed appropriately and rigorously?

Reviewer #2: Yes

4. Have the authors made all data underlying the findings in their manuscript fully available?

Reviewer #2: Yes

5. Is the manuscript presented in an intelligible fashion and written in standard English?

Reviewer #2: Yes

Reviewer #2: (No Response)

.

Reviewer #2: **Yes:** Chris LyalChris LyalChris LyalChris Lyal

---

## [Editor Report · Acceptance letter]

PONE-D-25-45522R1

PLOS One

Dear Dr. Groom,

I'm pleased to inform you that your manuscript has been deemed suitable for publication in PLOS One. Congratulations! Your manuscript is now being handed over to our production team.

Kind regards,

on behalf of

Dr. Daniel de Paiva Silva

Academic Editor

PLOS One